# Non-Invasive Evaluation of Intradiscal Deformation during Axial Loading of the Spine Using Deformation-Field Magnetic Resonance Imaging: A Potential Tool for Micro-Instability Measurements

**DOI:** 10.3390/jcm11164665

**Published:** 2022-08-10

**Authors:** Frida Johansson, Zainab Sirat, Hanna Hebelka, Helena Brisby, Fredrik Nordström, Kerstin Lagerstrand

**Affiliations:** 1Institute of Clinical Sciences, Sahlgrenska Academy, University of Gothenburg, 40530 Gothenburg, Sweden; 2Department of Medical Physics and Biomedical Engineering, Sahlgrenska University Hospital, 41345 Gothenburg, Sweden; 3Department of Radiology, Sahlgrenska University Hospital, 41345 Gothenburg, Sweden; 4Department of Orthopaedics, Sahlgrenska University Hospital, 41345 Gothenburg, Sweden

**Keywords:** MRI, low back pain, image registration, disc degeneration, disc deformation

## Abstract

Degeneration alters the structural components of the disc and its mechanical behavior. Understanding this pathophysiological process is of great importance, as it may lead to back pain. However, non-invasive methods to characterize the disc mechanics in vivo are lacking. Here, a potential method for measurements of the intradiscal deformation under stress is presented. The method utilizes a standard MRI protocol, commercial loading equipment, and registration software. The lumbar spine (L1/L2–L5/S1) of 36 human subjects was imaged with and without axial loading of the spine. The resulting images were registered, and changes in the images during the registration were displayed pixel-by-pixel to visualize the internal deformation of the disc. The degeneration grade, disc height, disc angle and tilt angle were determined and correlated with the deformation using multivariate regression analysis. The largest deformation was found at the lower lumbar spine, and differences in regional behaviors between individual discs were found. Weak to moderate correlations between the deformation and different disc characteristics were found, where the degeneration grade and tilt angle were the main contributing factors. To conclude, the image-based method offers a potential tool to study the pathophysiological process of the disc.

## 1. Introduction

The disc is a unique structure that allows the mobility of the spinal column and responds to changes in load by undergoing an instantaneous elastic deformation [1,2]. The composition of the disc, with an inner, more gel-like structure (nucleus pulposus) surrounded by a fibrocartilaginous circle (annulus fibrosus, is an ingenious construction distributing the compressive load within the spinal segments. Degenerative changes may alter the structural integrity and the mechanical response of the disc, influencing the overall motion of the spinal segments [3], as well as transferring more of the load bearing from the nucleus pulposus to the annulus fibrosus. Disc degeneration may at some stage lead to micro-instability of the spinal segment, which is believed to be associated with low back pain (LBP) [4]; one of the most prevalent disorders and the leading cause of years lived with disability in Western countries [5]. Given the unclear etiology of degeneration-related low back pain and the lack of an accepted disease model, comprehensive treatment remains elusive [3]. Further, the lack of well-defined biomechanical benchmarks concerning the in vivo load distribution and deformation patterns may have hindered the successful translation of promising surgical procedures into clinical reality [6].

While ex vivo experiments can be used to systematically study specific questions, they are not adequate surrogates for in vivo studies, as they cannot capture the complexity of the biomechanics of the discs. The development of methods for evaluation of the disc function in vivo may yield greater insight into the pathophysiological mechanisms of the disc during degeneration and may assist in the selection procedure for different treatment methods. Furthermore, the response to physiological weight-bearing loads may lay a foundation for identifying pathological deformation patterns in the discs that may contribute to pain; however, such measurements remain largely undocumented in vivo.

Magnetic resonance imaging (MRI), providing excellent image contrast and sub-millimeter resolution, is an established diagnostic tool in clinical practice. MRI measurements are of particular interest in load-bearing tissues such as the intervertebral discs, and when performed during axial loading of the spine, may permit characterization of tissues of biomechanical importance where pathological conditions could lead to micro-instability.

A previous study has shown that nonrigid image registration provides an accurate mapping of large and heterogeneous tissue deformations [7] and, as such, could be applied to discs. Here, a method is proposed that relies on axial loading during conventional MRI in combination with nonrigid image registration to calculate intradiscal deformation changes in vivo.

The present work aimed to study the feasibility of the proposed method for evaluation of the intradiscal deformation in vivo and associate the estimated intradiscal deformation with different disc characteristics.

## 2. Materials and Methods

MR image sets of the lumbar spine (L1/L2 to L5/S1) of 36 subjects (24 LBP patients and 12 healthy volunteers; mean age 38 years; range 25–69 years; 18 males), acquired in the morning between 8–12 AM, were used in the present study. The image sets, containing stacks of images acquired during MRI with and without axial loading of the spine, have been used in previously published work to investigate how the MR signal changes with disc degeneration [8,9,10].

All subjects gave informed consent. The study was performed according to the Declaration of Helsinki and approved by the Regional Ethics Review Board.

### 2.1. Image Acquisition

The MR images were acquired using a 1.5 T MRI scanner (Siemens Magnetom Aera, Erlangen, Germany). T1-weighted (slice thickness = 3.5 mm, field of view = 300 mm × 300 mm, TR = 480 ms, TE = 9 ms) and T2-weighted images (slice thickness = 3.5 mm, field of view = 300 mm × 300 mm, TR = 3500 ms, TE = 95 ms) were acquired in both conventional supine position and in supine position with axial compressive load of the spine.

### 2.2. Axial Compression

A commercial compression device (DynaWell Diagnostics Inc., Las Vegas, NV, USA), enabling axial loading of the spine during the examination of patients in the supine position, was used. The device, which consists of a spinal compression harness, adjustable side straps, and compression monitoring device, enabled standardized loading of the spine in the length axis between the head and feet. To further standardize the examination setup, all subjects were examined with 50% load of the body weight (similar to upright standing) and a customized pillow under the lumbar back to prevent flexion during loading.

### 2.3. Disc Characteristics

The degree of disc degeneration was characterized according to the classical Pfirrmann scheme with gradings from 1 to 5 [11], utilizing the T2-weighted images. A senior radiologist with more than 15 years of experience (H.H) performed the Pfirrmann grading, which has shown high intra- and moderate inter-observer agreement [9]. Several other disc characteristics were extracted from the T1-weighted images acquired during loading (Figure 1). The disc height was measured at the highest part of the disc, perpendicular to the disc plane. To overcome the influence of external factors on the tilt angle, a reference plane within each individual was chosen: the superior endplate of L1 (approximately perpendicular to the compressive force). Thus, the tilt angle was measured as the angle between the superior endplate of L1 and the inferior endplate of the actual disc. The disc angle was measured as the angle between the superior endplate of the lower vertebra and the inferior endplate of the upper vertebra. The angle and height measurements were performed manually by a Ph.D. student. After 6 months, the Ph.D. student and the senior radiologist repeated the angle and height measurements independently on 30% of the data set, blinded to the original results. To verify that the measured deformation displayed a realistic distribution over the disc, it was correlated to the change in the disc angle. That is, a large change in the disc angle after loading was expected to be associated with a large difference in the deformation value between subregion 1 and 5. The sub-analysis was performed in all discs at L4–L5, where the largest deformation effect was expected.

### 2.4. Measurement of the Intradiscal Deformation

The intradiscal deformation was determined as follows: The T2-weighted image that was acquired during loading of the spine was non-rigidly registered to the corresponding image that was acquired without loading of the spine. The deformation field, which describes the deformation of the image during the registration, was used to calculate the Jacobian determinant for each voxel. The spatial distribution of the determinants in the voxels within the disc, representing the intradiscal deformation distribution, was displayed as a heatmap (Figure 2). To further analyze the intradiscal deformation, all intervertebral discs in each heatmap were delineated, and the mean deformation was calculated. From the whole volume of the discs, the three mid-sagittal slices of the discs were selected to calculate the mean deformation in five evenly distributed subregions, from anterior (region 1) to posterior (region 5). The image registration, deformation analysis, and disc segmentation are described in detail below.

### 2.5. The Image Registration

The well-validated registration software Elastix (Version 5.0.0, University Medical Center, Utrecht, The Netherlands) [12] was used to register the stacks of the sagittal T2-weighted images, acquired with and without loading of the spine. To facilitate automatic and advanced image analysis, the MICE Toolkit (Version 1.1.0, Nonpi Medical AB, Umeå, Sweden) was used as a graphical user interface for Elastix. The default registration settings of Elastix were adjusted to optimize the registration quality for the specific research question (Appendix A). After increasing the matrix resolution to 1 mm isotropic voxels using linear interpolation, the registration was performed in two steps: (1) rigid Euler transformation and (2) non-rigid B-spline transformation (Figure 2), resulting in no negative Jacobian determinants. The MR images of the loaded spine were transformed into the corresponding image space of the unloaded spine. The transformed image was then resampled back to the original matrix dimensions of the image without loading of the spine.

The quality of the image registration was ensured by determining two commonly used similarity metrics: the Dice similarity coefficient (0.85 ± 0.06) and the Jaccard coefficient (0.74 ± 0.08). For that purpose, the disc segmentations were registered using the same transformation as in the registration of the corresponding MR images for each individual.

### 2.6. The Intradiscal Deformation Analysis

To determine the intradiscal deformation, the deformation field of the image during the registration was calculated. Then, the Jacobian matrix (J) was calculated through partial derivatives of each element of the deformation field (f) as follows:$$J=\left[\begin{array}{ccc} \frac{\partial f_{1}}{\partial x_{1}} & \cdots & \frac{\partial f_{1}}{\partial x_{n}}\\ \vdots & \ddots & \vdots \\ \frac{\partial f_{m}}{\partial x_{1}} & \cdots & \frac{\partial f_{m}}{\partial x_{n}} \end{array}\right]$$

Finally, the determinant of the Jacobian matrix (|J|) was calculated voxel by voxel, representing the intradiscal deformation, where a value less than one represented expansion and a value greater than one represented compression when the axial load was applied.

### 2.7. The Disc Segmentation

An in-house trained convolutional neural network, implemented in the Dragonfly software (Version 2020.1 for Microsoft Windows, Object Research Systems (ORS) Inc., Montreal, QC, Canada, 2020) was used to automatically segment the lumbar discs on all sagittal T2-weighted images where the disc was clearly visible. High agreement with ground truth was established for the current segmentation method (Dice similarity coefficient: 0.89–0.93 depending on the visibility and geometrical complexity of the disc).

### 2.8. Statistical Analysis

SPSS Statistics version 17 (IBM, Armonk, NY, USA) was used for all statistical analysis, and if not otherwise stated, the deformation is presented as the grouped mean ± standard deviation of the mean Jacobian determinant. The Kruskal–Wallis H test was performed to assess the difference in deformation between discs of different spine levels and between different subregions and slices with a significance level of *p* < 0.05. Pearson’s correlation coefficient (R) was used to calculate the correlation between all deformation measures and disc characteristics including the spinal level. Regression analysis was performed to generate both univariable and multivariable linear models. With the deformation parameters as dependent variables and the disc characteristics (Pfirrmann grade, disc height, tilt angle, and disc angle) as independent variables, the correlation between deformation parameters and disc characteristics was evaluated using the interpretation of the correlation coefficient R developed by Chan [13]. A reliability test for inter- and intra-rater agreement of the disc characteristics was performed by estimating the intraclass correlation coefficient (ICC) with 95% confidence interval using the single measure, absolute agreement, 2-way mixed model.

## 3. Results

### 3.1. General

All subjects tolerated the compression well and were successfully examined with MRI, and none of the subjects were excluded from the study, resulting in a total of 180 discs. Further, the imaging quality of the MRI images was satisfactory, and the proposed method was found to display the intradiscal deformation with both high contrast and spatial resolution (Figure 3 and Figure 4). The inter- and intra-observer agreement was found to be excellent for the tilt angle (inter: ICC = 0.99, intra: ICC = 0.99), good for the disc angle (inter: ICC = 0.76, intra: ICC = 0.82) and moderate for the disc height measurement (inter: ICC = 0.72, intra: ICC = 0.68).

### 3.2. Correlations between Disc Characteristics

The Pfirrmann grade was not found to correlate with the tilt angle or disc angle. Of all extracted disc characteristics, only the tilt angle and disc angle were shown to be associated, however, weakly (R = 0.5, *p* < 0.001). The correlation between the spine level and the tilt angle was high (R = −0.9, *p* < 0.001), while weak correlations were found between the spine level and the disc angle (R = −0.5, *p* < 0.001) and the spine level and Pfirrmann grade (R = −0.4, *p* < 0.001). Furthermore, the disc height did not differ significantly between levels.

### 3.3. Intradiscal Deformation Correlations

In general, the intradiscal deformation varied with the spine level, where larger deformation (compression) was found in discs at the lower levels of the spine (Figure 3 and Figure 4). The difference in the intradiscal deformation between different spine levels was statistically verified (mean range: 1.12 (L5–S1) to 0.928 (L1–L2), *p* < 0.001). Further, the intradiscal deformation displayed overall a regional variation both in the anterior–posterior and left–right directions (*p* = 0.007; Figure 4).

When including all extracted disc parameters, i.e., the Pfirrmann grade, disc height, tilt angle, and disc angle, in a multivariate linear regression model, a weak to moderate correlation was found for the whole disc as well as for the central subregions (Figure 5B). As shown by the β-values, discs with larger tilt angles and higher Pfirrmann grades displayed more deformation. The largest change was found at the center and at the anterior part of the disc, as well as in the rightmost slices. Backward elimination of the dependent variables in the regression model revealed that the tilt angle was the strongest correlate for the intradiscal deformation with larger deformation values at larger tilt angles (Figure 5). Degeneration measured in terms of Pfirrmann grade was also found to be a correlate for the intradiscal deformation (Figure 5C). The absolute change in deformation between discs with different Pfirrmann grades, as calculated from the β-values, was found to be slightly lower than 0.1. This could be compared with the change in deformation between discs with different tilt angles, which could be as high as 0.2. After correcting for the influence of the spine level, the intradiscal deformation showed a significant correlation with the tilt angle and disc height (Appendix A). For the caudal discs, the tilt angle was the strongest correlate (β = 0.003–0.004, *p* = 0.001–0.03), and for the cranial discs, the disc height was the strongest (β = −0.02–−0.01, *p* ≤ 0.001–0.05).

The changes in the disc angle in the discs at L4-L5 were found to be weakly correlated to the difference in the deformation value at subregion 1 and 5 (R = 0.4; *p* = 0.04), verifying the expected behavior of the deformation over the discs.

## 4. Discussion

Prior attempts to quantify the load–deformation response of the disc most often rely on in vitro preparations and, as such, display non-physiological behaviors. The proposed method, which utilizes the deformation field generated during the registration of conventional MR images taken with and without axial loading, showed high feasibility in characterizing the intradiscal deformation in vivo. With this clinically applicable concept, the deformation could not only be visualized, but also quantified to enable longitudinal follow-up studies for improved understanding of the pathophysiological mechanisms of the disc, which is of importance to understand, e.g., the degenerative process. Further studies are needed to confirm the present findings, establish normal baseline values and validate the diagnostic performance of the method for evaluation of pathological disorders.

Previous work has shown that axial loading during MRI introduces changes in the disc signal [14] and that the size of the signal alteration differs between symptomatic LBP patients and asymptomatic controls [15,16]. It has been suggested that these findings most probably reflect a redistribution of the water in the nucleus pulposus. Direct measurements of the deformation of the disc may display other functional properties, e.g., collagen and proteoglycan content/relationship. Micro-instability of the disc, defined as “active discopathy” by Nguyen et al. [17], has been suggested to configure the first phase of the degenerative cascade in the spine [18] and to be involved in degenerative disorders such as LBP [19]. However, the condition poses a diagnostic challenge, and its association with LBP is a subject of debate. Hence, there is a need to establish new diagnostic methods that can display the ability to withstand compression and display the micro-stability or micro-instability of the disc.

While in vivo measurements of intradiscal deformation may shed light on disc pathologies and the site-specific nature of disc tears and herniations [20], they pose large technical challenges and, as such, only a few studies have presented methods for human applications. Byrne et al. have presented a method based on point-wise mapping to quantify the biomechanical properties of the disc [1]. However, their method relies on multiple radiological examinations utilizing ionizing radiation and is thereby unsuitable for longitudinal follow-up studies. In a small proof-of-concept study, Menon et al. proposed a non-ionizing method based on MRI combined with optical flow analysis [21]. Due to the intrinsically higher contrast of the MRI technique, internal structures of the disc could be better depicted and, as such, the method seems more promising for quantification of the intradiscal deformation. However, unlike the method presented here, their method does not rely on conventional MRI scanning or on validated commercial software. This makes the method less suitable for clinical use. Additionally, the optical flow algorithm may introduce errors that can propagate into the calculated results.

Similarly to our study, Menon et al. showed that the deformation during loading varies depending on the position of the disc segment, where the largest compressive deformation was seen at the lowest level of the spine. They also showed, in consensus with us, highly complex distribution of the deformation values over the discs, emphasizing the need for methods with the ability to map these disc changes pixel-by-pixel. While global measurements provide important data [22], characterization of the deformation of the inner portion of the disc may reveal dysfunctional behaviors at an earlier phase of the degenerative cascade. The deformation pattern may contribute to building a stronger foundation of knowledge of why patients with some spine types, e.g., Roussouly type 1 [23], might be more prone to develop disc degeneration and display disc injuries [24,25].

The feasibility of the present method to accurately map the internal deformation within the disc, as previously shown ex vivo [7], was further strengthened by our findings, displaying a strong association between changes in the deformation over the disc and the disc angle. While no gold standard exists for measurements of the intradiscal deformation in vivo, our findings were found to agree with previous studies, demonstrating, for instance, that discs with more degeneration display larger compressive deformation [26,27,28,29]. Degeneration is known to change with age and progressively alter the tissue composition and structure of the disc, including proteoglycan loss and a corresponding reduction in osmotic pressure. This may explain the decrease in the compression stiffness with higher degeneration grades [30]. Another interesting finding of the study was the presence of asymmetries in the deformation over the disc in the left–right direction. In a confirming ex vivo study, Chan et al. visualized such asymmetrical right–left deformation behaviors in specimens of disc segments during the application of an axial compression load. Byrne et al. [1] presented similar results in vivo and argued that the behavior might be the effect of an intrinsic variation in the disc geometry, including the shape and orientation of the cartilage endplates. This, in turn, may have introduced a variation in the distribution of the load over the disc, shifting the position of the nucleus slightly off-center [26]. At the weight-bearing condition during upright standing position, or as simulated here during the application of the compression device, differences in leg length and muscle strength may modify the spinal load. Furthermore, scoliosis, which represents an abnormal lateral curvature of the spine, may also introduce large variations in the load over the discs and, thus, induce asymmetric intradiscal deformations that may change with the progression of the disorder. With the proposed method, underlying mechanical properties of the disc could be characterized to improve the understanding of the process and possibly at an earlier stage to detect risk behaviors in patients with degenerative scoliosis. The present study further showed that the intradiscal deformation depends on the disc angle and the angulation of the disc in relation to the loading force, here indirectly measured in terms of the tilt angle. Even though no study has previously demonstrated this effect, we expected to find differences in deformation behaviors between discs with diverse morphologies due to the intrinsic variation in the loading distribution over the disc. Future studies should investigate the relevance of the loading behavior on the degenerative process and elucidate how the intradiscal deformation varies with the inherent changes in the disc matrix structure.

The present cohort consisted both of patients with LBP and healthy individuals. Based on the limited sample size, however, no attempt to analyze these two groups separately was made here. With larger sample sizes, we believe that analyses in relation to individual parameters, such as age, BMI and gender, as well as pain profile, functional status and exercise profile, will be of great interest to study in relation to intradiscal deformation.

### Limitations

The compression device, which has been extensively validated in various research studies and in clinical practice [15,16,31], allowed MRI scanning during the stress-loaded condition in a motionless supine position. However, anatomic structures, such as leg length and muscle strength, may have modified the applied load using this device. As such, in-between subject variations may have been introduced. On the other hand, the value of the present method is mainly its ability to evaluate the mechanical function of the discs in humans during realistic loading conditions and, as such, the anatomic structures of the individuals should influence the loading distribution.

Furthermore, the intradiscal deformation was determined from images acquired only at two different loading states, that is, at a relaxed state in the supine position and at a state that simulated the loading of the spine in standing upright position. Even though axial loading during supine MRI has been shown to be comparable to standing MRI [31], standing MRI enables the evaluation of intradiscal deformation also under other interesting loading conditions with a larger degree of freedom.

The gross grading scheme of disc degeneration used in this study, i.e., the Pfirrmann grade, which has been shown to display a weak association with LBP, was here shown to correlate only weakly with the deformation of the disc. The weak correlation might reflect the limitation in the Pfirrmann grading scheme to characterize important properties of the disc related to future pain, but this might be too early to speculate.

## 5. Conclusions

Findings suggest that the proposed image-based method offers a new way of identifying the internal deformation of the disc under stress in vivo and can non-invasively quantify important biomechanical properties of the disc associated with disc degeneration as well as the disc and tilt angle. With this clinically applicable concept, utilizing axial loading during conventional MRI in combination with image registration to calculate the deformation field, the understanding of the pathophysiological mechanism of the disc might be improved.

## Figures and Tables

**Figure 1 jcm-11-04665-f001:**
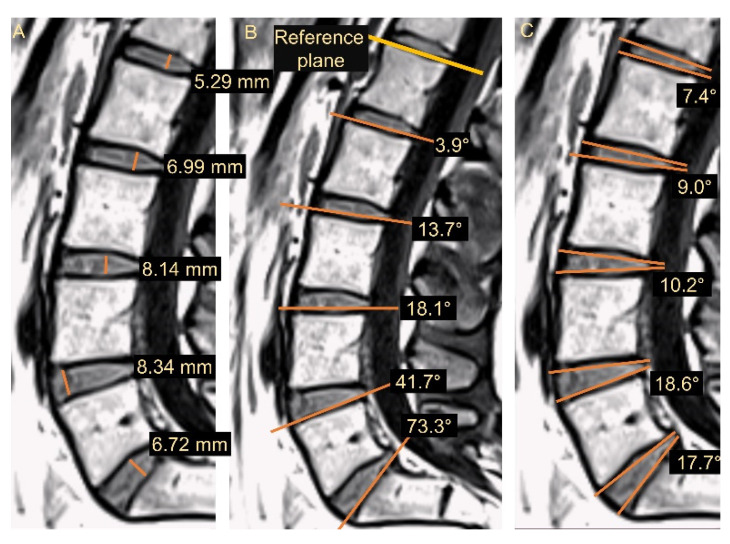
Illustration of the disc measures estimated in the study. (**A**) The disc height, (**B**) the tilt angle for L5–S1, and (**C**) disc angle for all IVDs.

**Figure 2 jcm-11-04665-f002:**
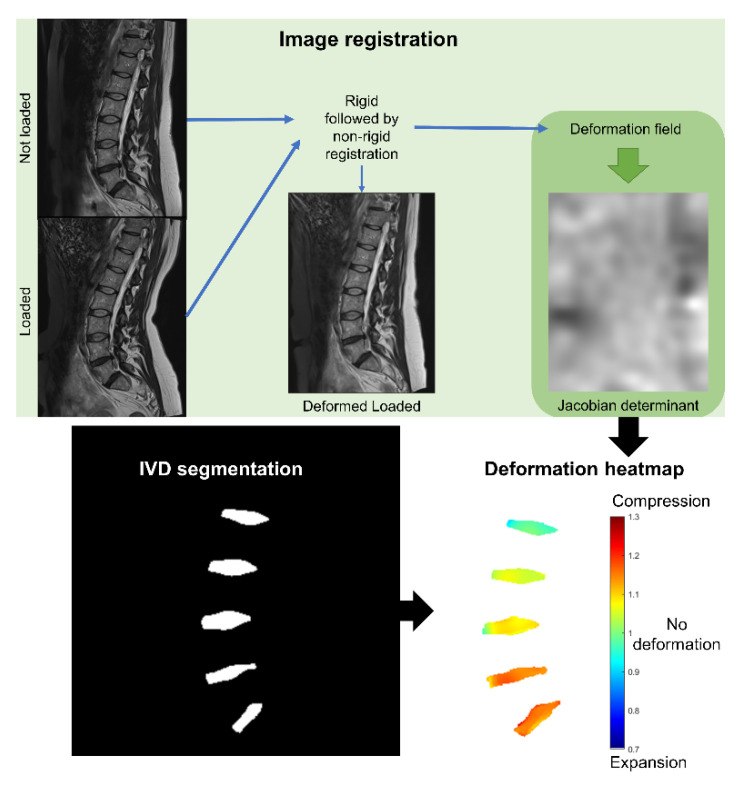
Schematic illustration of the method to measure the intradiscal deformation. The T2-weighted image of the loaded lumbar spine was rigidly and then non-rigidly transformed to the space of the MR image of the unloaded spine. The deformation field, retrieved from the registration, was used to calculate the Jacobian determinant. The Jacobian determinant in the IVDs, which represented the intradiscal deformation, was displayed in detail as heatmaps, where the IVDs were selected automatically using automatic segmentation of the IVDs.

**Figure 3 jcm-11-04665-f003:**
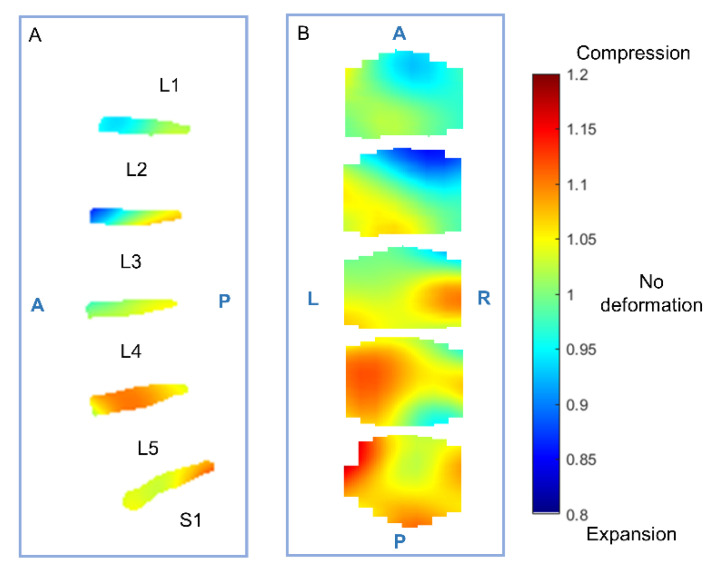
Heatmaps of the intradiscal deformation. Displayed as a sagittal projection (**A**), and a head–feet projection (**B**), where a Jacobian determinant larger than 1 represents compression and smaller than 1 represents expansion. Larger compression was found posteriorly in the lower lumbar spine, and generally, the compression showed a left asymmetric distribution pattern.

**Figure 4 jcm-11-04665-f004:**
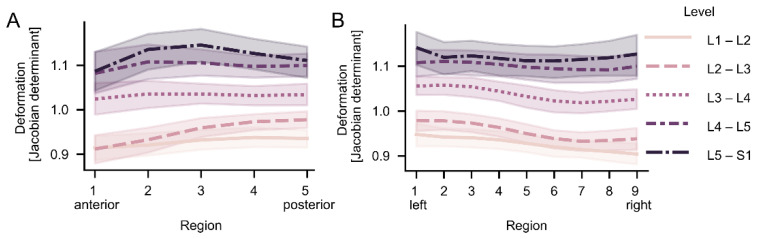
Regional variation of the deformation for different spine levels. Mean intradiscal deformation with (<1 = expansion, >1 = compression) as a function of the spine level for (**A**) the nine mid slices of the disc and (**B**) the five anterior to posterior regions of the discs.

**Figure 5 jcm-11-04665-f005:**
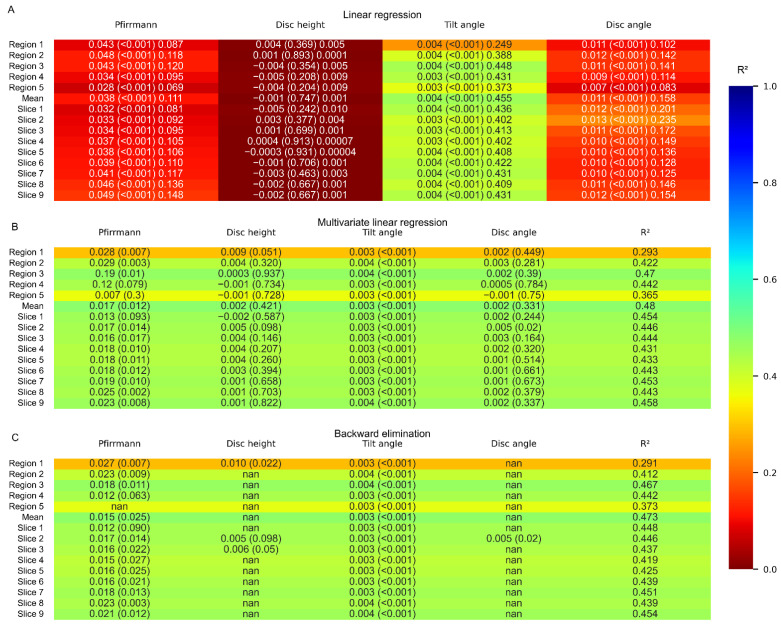
Linear regression analysis. (**A**) Univariate linear regression, (**B**) multivariate linear regression and (**C**) multivariate linear regression with backward elimination colored according to the coefficient of determination for the regression between the deformation and the disc characteristics. The cells contain β (*p*-value) R^2^ for A and β (*p*-value) with R^2^ in a separate column for B and C. Region 1 = anterior, Region 5 = posterior, Slice 1 = left, and Slice 9 = right.

## Data Availability

Data supporting the reported findings are available on request.

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
