# Peer review of "Non-Invasive Evaluation of Intradiscal Deformation during Axial Loading of the Spine Using Deformation-Field Magnetic Resonance Imaging: A Potential Tool for Micro-Instability Measurements"

_jcm, 2022, doi:10.3390/jcm11164665_

Round 1

Reviewer 1 Report

I found this idea of measuring in vivo the dynamic deformation of the discs very interesting. However there are some issues that probably would improve the paper:

1) Is the sample composed of healthy individuals? Was low back pain considered as exclusion or inclusion criteria? What about their BMI, moment of the day (morning or afternoon), state od hydration and so on? (factors that could influence the behaviour of the discs)

2)The Pfirrmann grading was performed by an experienced radiologist (and later they added a supervised PhD student) I think it is mandatory to perform an inter and intraobserver reliability test (the aforementioned grading has it, but, as any measuring instrument, it is recomended to test the reliability of the measures done by the observers)

3) In the conclusion, the authors say that "discs with more degeneration display larger compressive deformation" But I couldn´t find any mention about how they assess discal degeneration or when did they correlate discal degeneration with compressive deformation

4) The same about the Roussouly spine type, where is this correlation shown?

In short, the article is interesting, but needs major revision

Author Response

Dear reviewer,

We thank you for taking the time to review our manuscript “Non-invasive evaluation of intradiscal deformation during axial loading of the spine using deformation-field magnetic resonance imaging: A potential tool for micro-instability measurements” and for your valuable and relevant suggestions to improve our work. Based on your suggestions we have altered the manuscript and believe that it has been greatly improved. We have attached a point-to-point reply to your comments. Changes in the manuscript have been highlighted in gray.

Reviewer 1:

Comment 1: I found this idea of measuring in vivo the dynamic deformation of the discs very interesting. However there are some issues that probably would improve the paper:

Is the sample composed of healthy individuals? Was low back pain considered as exclusion or inclusion criteria? What about their BMI, moment of the day (morning or afternoon), state od hydration and so on? (factors that could influence the behavior of the discs)

Answer 1: The cohort consisted of 24 LBP-patients and 12 healthy volunteers. It has now been clarified in the Method section at page 2, line 70-71. We have also added a section at page 9, line 332-337 to discuss this.

In the present study, the time point of the examination had been standardized; all subjects were examined between 8 and 12 AM. This missing information has been included in the Method section at page 2, line 71-72.

Also other factors, e.g. BMI and gender, might affect the deformation of the disc. Due to the reduction in statistical power following multiple testings, however, these factors were not evaluated in the proof-of-concept study but are planned in a future large cohort study (page 9, line 332-337).

Comment 2: The Pfirrmann grading was performed by an experienced radiologist (and later they added a supervised PhD student) I think it is mandatory to perform an inter and intraobserver reliability test (the aforementioned grading has it, but, as any measuring instrument, it is recomended to test the reliability of the measures done by the observers)

Answer 2: In the present study, inter- and intra-observer reliability tests of the measures have been performed by the observers. We realize that that the manuscript has been unclear on this point and thank you for highlighting this. The manuscript has been adjusted accordingly at page 3, line 104-107.

The Pfirrmann grading was done by experienced radiologists, resulting in high intra- and moderate inter-observer agreement (page 3, line 93-94).

The PhD student did the angle and height measurements, including the intra-observer reliability test (page 3, line 104-107). For the inter-observer reliability test, an experienced radiologist also performed the angle and height measurements. The results can be found at page 5, line 195-198.

Comment 3: In the conclusion, the authors say that "discs with more degeneration display larger compressive deformation" But I couldn´t find any mention about how they assess discal degeneration or when did they correlate discal degeneration with compressive deformation

Answer 3: We used the classical Pfirrmann grading system to assess the degeneration of the disc for correlating with compressive deformation (page 3, line 93-94). Findings are presented in the Result section at page 7, line 232-233.

Comment 4: The same about the Roussouly spine type, where is this correlation shown?

Answer 4: In the present study, we discussed the Roussouly spine type as a factor that could influence the behaviour of the disc and, as such, should be added in future work. Due to the lack of standing X-ray, however, we did not evaluate the Roussouly spine type.

Comment 5: In short, the article is interesting, but needs major revision

Reviewer 2 Report

The present study aims to non-invasively evaluate the intradiscal deformation during axial loading with MRI.  

The study is very interesting, and it is well conducted, some minor points should be specified to improve the manuscript:

- The authors should clarify if 36 patients were healthy volunteers or patients with low back pain

- Patients with Pfirmann grade 6 or 7 or 8 would be excluded if the Modified Pfirrmann grading system is used?

- Is it possible to evaluate discs deformation related to sagittal balance of the patients? Is there a role of pelvic tilt and sacral slope? 

Author Response

Dear reviewer,

We thank you for taking the time to review our manuscript “Non-invasive evaluation of intradiscal deformation during axial loading of the spine using deformation-field magnetic resonance imaging: A potential tool for micro-instability measurements” and for your valuable and relevant suggestions to improve our work. Based on your suggestions we have altered our manuscript and believe that it has been improved. We have attached a reply point-to-point to your comments. Changes to the manuscript have been highlighted in gray.

Reviewer 2:

Comment 1: The present study aims to non-invasively evaluate the intradiscal deformation during axial loading with MRI. 

The study is very interesting, and it is well conducted, some minor points should be specified to improve the manuscript:

- The authors should clarify if 36 patients were healthy volunteers or patients with low back pain

Answer 1: The cohort consisted of 24 LBP-patients and 12 healthy volunteers. It has now been clarified in the Method section at page 2, line 70-71. We have also added a section at page 9, line 332-337 to discuss this.

Comment 2: - Patients with Pfirrmann grade 6 or 7 or 8 would be excluded if the Modified Pfirrmann grading system is used?

Answer 2: The modified Pfirrmann grading system was not used, instead we used the classical grading system from 1 to 5. We realized that this information has been vague and has now been enhanced in the Method section at page 3, line 93-94. 

Comment 3: - Is it possible to evaluate discs deformation related to sagittal balance of the patients? Is there a role of pelvic tilt and sacral slope?

Answer 3: This is a very interesting point, which we cannot fully answer, even if the relation found between tilt angle of the disc and deformation suggests that spine type and possibly sagittal balance might play a role. We are not aware of any study where this has been studied and since we do not have standing x-rays on most of the patients in the cohort, we cannot evaluate this further in the present work. This is, however, something we think is of interest to explore in future studies.

Reviewer 3 Report

It is of great significance to the spine surgeon to examine the spine with axial pressure. I have never seen your paper's methodology before, but I found it to be an objective method and a scientific means of ensuring statistical reliability.

I have two questions.

1. I can imagine that the  lordotic alignment of the lumbar spine brings about differences in stress distribution in the anteroposterior direction of the intervertebral discs.  However, what is the influence of the difference in stress distribution in the left-right direction? I would like to know the author's thoughts on whether it is an error in the validation method or whether cases with scoliosis were included.

2. It seems to me that the inclusion of patients with kyphosis or scoliosis in the spinal alignment would greatly affect the validation results. What do you think about the cases included in this study?

・The introductory section (page 1, line 38) 

The formal description of AF should be presented. (It is described 4-5 lines earlier, but it is difficult to understand. In addition, since AFs do not appear in large numbers, we feel it is not necessary to provide an abbreviated description).

Author Response

Dear reviewer,

We thank you for taking the time to review our manuscript “Non-invasive evaluation of intradiscal deformation during axial loading of the spine using deformation-field magnetic resonance imaging: A potential tool for micro-instability measurements” and for your valuable and relevant suggestions to improve our work. Based on your suggestions we have altered our manuscript and believe that it has been improved. We have attached a reply point-to-point to your comments. Changes to the manuscript have been highlighted in gray.

Reviewer 3:

Comments and Suggestions for Authors

It is of great significance to the spine surgeon to examine the spine with axial pressure. I have never seen your paper's methodology before, but I found it to be an objective method and a scientific means of ensuring statistical reliability.

I have two questions.

  1. I can imagine that the lordotic alignment of the lumbar spine brings about differences in stress distribution in the anteroposterior direction of the intervertebral discs. However, what is the influence of the difference in stress distribution in the left-right direction? I would like to know the author's thoughts on whether it is an error in the validation method or whether cases with scoliosis were included.

Answer 1: This is a very interesting point, which we cannot fully answer. A left-right distribution of the deformation was found. However, no subjects with scoliosis were included in the study cohort that would have explained this finding. Other groups have displayed similar deformation distributions in the lumbar discs, using ex vivo samples or different validation methods, suggesting that the effect is real and not an error in the validation method. As discussed in the manuscript, several factors can have and influence on the left right distribution, for example disc height and sagittal balance. However, this is only speculations and need further evaluation.

  1. It seems to me that the inclusion of patients with kyphosis or scoliosis in the spinal alignment would greatly affect the validation results. What do you think about the cases included in this study?

Answer 2: The inclusion of cases with kyphosis or scoliosis in the spinal alignment would most probably have affected the validation results and are a given application for the proposed method. None of the cases in the present study, however, had kyphosis or scoliosis, as verified by the experienced radiologist.

The introductory section (page 1, line 38). The formal description of AF should be presented. (It is described 4-5 lines earlier, but it is difficult to understand. In addition, since AFs do not appear in large numbers, we feel it is not necessary to provide an abbreviated description).

Answer: Thank you for pointing this out. At your recommendation, we have adjusted the description of AF at page 1, line 34. We also agree that there is no need to provide an abbreviated description to AF and has adjusted this accordingly throughout the manuscript. 

Reviewer 4 Report

Here we raise the following questions, which we hope the authors will refer to: 1. There is a large age difference in case selection, and there is a lack of systematic discussion of this difference. 2. Although the measurement of the data is guided by a senior person, is there any error in the measurement by a single person?

Author Response

Dear reviewer,

We thank you for taking the time to review our manuscript “Non-invasive evaluation of intradiscal deformation during axial loading of the spine using deformation-field magnetic resonance imaging: A potential tool for micro-instability measurements” and for your valuable and relevant suggestions to improve our work. Based on your suggestions we have altered our manuscript and believe that it has been improved. We have attached a reply point-to-point to your comments. Changes to the manuscript have been highlighted in gray.

Reviewer 4:

Comments and Suggestions for Authors

Here we raise the following questions, which we hope the authors will refer to: 

  1. There is a large age difference in case selection, and there is a lack of systematic discussion of this difference. 

Answer 1: Thank you for pointing this out. The age is a factor that should have an influence on the deformation, since degeneration is known to change with age and progressively alter the tissue composition and structure of the disc. We have modified the text in the Discussion section at page 9, line 305-307 to describe this clearer.

Based on the limited sample size, no attempt to stratify on age separately was performed. In the present study, we stratified on disc degeneration according to the classical Pfirrmann grading scheme, which should include some of the changes seen with age.

Your comment raised the awareness that we needed to improve the manuscript by also including other factors that should be of importance in future large cohort evaluations. Please see page 9, line 332-337.

  1. Although the measurement of the data is guided by a senior person, is there any error in the measurement by a single person? 

Answer 2: The measurements were performed by two observers and both inter- and intra-observer agreements were determined. We realize that the manuscript has been unclear on this point and thank you for highlighting this. The manuscript has been adjusted accordingly at page 3, line 104-107.
